# Bidirectional Learning for Offline Infinite-width Model-based Optimization

**Can (Sam) Chen[1]\*, Yingxue Zhang[2], Jie Fu[3], Xue Liu[1], Mark Coates[1]**

[1]McGill University, [2] Huawei Noah's Ark Lab, [3] Beijing Academy of Artificial Intelligence
`can.chen@mail.mcgill.ca, yingxue.zhang@huawei.com,`
`fujie@baai.ac.cn, xueliu@cs.mcgill.ca, mark.coates@mcgill.ca`

## Abstract

In offline model-based optimization, we strive to maximize a black-box objective function by only leveraging a static dataset of designs and their scores. This problem setting arises in numerous fields including the design of materials, robots, DNA sequences, and proteins. Recent approaches train a deep neural network (DNN) on the static dataset to act as a proxy function, and then perform gradient ascent on the existing designs to obtain potentially high-scoring designs. This methodology frequently suffers from the out-of-distribution problem where the proxy function often returns poor designs. To mitigate this problem, we propose ***BiD**irectional learning for offline **I**nfinite-width model-based optimization* (**BDI**). BDI consists of two mappings: the forward mapping leverages the static dataset to predict the scores of the high-scoring designs, and the backward mapping leverages the high-scoring designs to predict the scores of the static dataset. The backward mapping, neglected in previous work, can distill more information from the static dataset into the high-scoring designs, which effectively mitigates the out-of-distribution problem. For a finite-width DNN model, the loss function of the backward mapping is intractable and only has an approximate form, which leads to a significant deterioration of the design quality. We thus adopt an infinite-width DNN model, and propose to employ the corresponding neural tangent kernel to yield a closed-form loss for more accurate design updates. Experiments on various tasks verify the effectiveness of BDI. The code is available here.

## 1 Introduction

Designing a new object or entity with desired properties is a fundamental problem in science and engineering [1]. From material design [2] to protein design [3, 4], many works rely on interactive access with the unknown objective function to propose new designs. However, in real-world scenarios, evaluation of the objective function is often expensive or dangerous [2–6], and thus it is more reasonable to assume that we only have access to a static (offline) dataset of designs and their scores. This setting is called offline model-based optimization. We aim to find a design to maximize the unknown objective function by only leveraging the static dataset.

A common approach to solve this problem is to train a deep neural network (DNN) model on the static dataset to act as a proxy function parameterized as $f_{\boldsymbol{\theta}}(\cdot)$. Then the new designs are obtained by performing gradient ascent on the existing designs with respect to $f_{\boldsymbol{\theta}}(\cdot)$. This approach is attractive because it can leverage the gradient information of the DNN model to obtain improved designs. Unfortunately, the trained DNN model is only valid near the training distribution and performs poorly

---

\*Corresponding author.

36th Conference on Neural Information Processing Systems (NeurIPS 2022).

outside of the distribution. More specifically, the designs generated by directly optimizing $f_{\theta}(\cdot)$ are scored with erroneously high values [7, 8]. As shown in Figure 1, the proxy function trained on the static dataset $p_{1,2,3}$ overestimates the ground truth objective function, and the seemingly high-scoring design $p_{\text{grad}}$ obtained by gradient ascent has a low ground truth score $p'_{\text{grad}}$.

Recent works [7–9] address the out-of-distribution problem by imposing effective inductive biases on the DNN model. The method in [8] learns a proxy function $f_{\theta}(\cdot)$ that lower bounds the ground truth scores on out-of-distribution designs. The approach in [9] bounds the distance between the proxy function and the objective function by normalized maximum likelihood. In [7], Yu et al. use a local smoothness prior to overcome the brittleness of the proxy function and thus avoid poor designs.

As demonstrated in Figure 1, these previous works try to better fit the proxy function to the ground truth function from a *model* perspective, and then obtain high-scoring designs by gradient ascent regarding the proxy function. In this paper, we investigate a method that does not focus on regularizing the model, but instead ensures that the proposed designs can be used to predict the available *data*. As illustrated in Figure 1, by ensuring the high-scoring design $p_{\text{ours}}$ can predict the scores of the static dataset $p_{1,2,3}$ (backward) and vice versa (forward), $p_{\text{ours}}$ distills more information from $p_{1,2,3}$, which makes $p_{\text{ours}}$ more aligned with $p_{1,2,3}$, leading to $p'_{\text{ours}}$ with a high ground truth score.

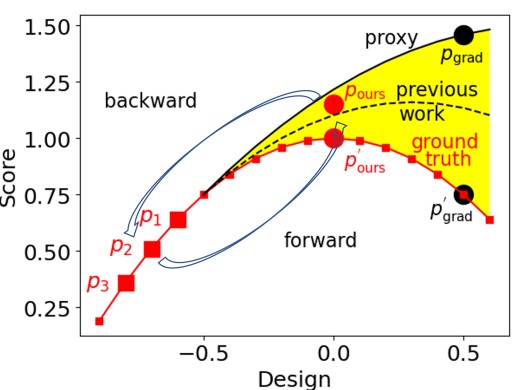

Figure 1: Illustration of motivation.

We propose ***BiD**irectional learning for offline **I**nfinite-width model-based optimization* (**BDI**) between the high-scoring designs and the static dataset (a.k.a. low-scoring designs). BDI has a forward mapping from low-scoring to high-scoring and a backward mapping from high-scoring to low-scoring. The backward mapping means the DNN model trained on the high-scoring designs is expected to predict the scores of the static dataset, and vice versa for the forward mapping. To compute the prediction losses, we need to know the scores of the high-scoring designs. Inspired by [10], where a predefined reward (score) is used for reinforcement learning, we set a predefined target score for the high-scoring designs. By minimizing the prediction losses, the bidirectional mapping distills the knowledge of the static dataset into the high-scoring designs. This ensures that the high-scoring designs are more aligned with the static dataset and thus effectively mitigates the out-of-distribution problem. For a finite-width DNN model, the loss function of the backward mapping is intractable and can only yield an approximate form, which leads to a significant deterioration of the design quality. We thus adopt a DNN model with infinite width, and propose to adopt the corresponding neural tangent kernel (NTK) [11, 12] to produce a closed-form loss function for more accurate design updates. This is discussed in Section 4.5. Experiments on multiple tasks in the design-bench [1] verify the effectiveness of BDI.

To summarize, our contributions are three-fold:

- We propose bidirectional learning between the high-scoring designs and the static dataset, which effectively mitigates the out-of-distribution problem.

- To enable a closed-form loss function, we adopt an infinite-width DNN model and introduce NTK into the bidirectional learning, which leads to better designs.

- We achieve state-of-the-art results on various tasks, which demonstrates the effectiveness of BDI.

## 2 Preliminaries

### 2.1 Offline Model-based Optimization

Offline model-based optimization is formally defined as:

$$\boldsymbol{x}^* = \arg\max_{\boldsymbol{x}} f(\boldsymbol{x}),\tag{1}$$

where the objective function $f(\boldsymbol{x})$ is unknown, but we have access to a size N dataset $\mathcal{D} = \{(\boldsymbol{x}_1, y_1)\}, \cdots, \{(\boldsymbol{x}_N, y_N)\}$, where $\boldsymbol{x}_i$ represents a certain design and $y_i$ denotes the design score. The design could, for example, be a drug, an aircraft or a robot morphology. Denote the feature dimension of a design $\boldsymbol{x}$ as D and then the static dataset $\mathcal{D}$ can also have a vector representation $(\boldsymbol{X}_l, \boldsymbol{y}_l)$ where $\boldsymbol{X}_l \in \mathcal{R}^{N \times D}, \boldsymbol{y}_l \in \mathcal{R}^N$.

A common approach is fitting a DNN model $f_{\boldsymbol{\theta}}(\cdot)$ with parameters $\boldsymbol{\theta}$ to the static dataset:

$$\boldsymbol{\theta}^* = \arg\min_{\boldsymbol{\theta}} \frac{1}{N} \sum_{i=1}^{N} (f_{\boldsymbol{\theta}}(\boldsymbol{x}_i) - y_i)^2 . \tag{2}$$

After that, the high-scoring design $\boldsymbol{x}^*$ can be obtained by optimizing $\boldsymbol{x}$ against the proxy function $f_{\boldsymbol{\theta}^*}(\boldsymbol{x})$ by gradient ascent steps:

$$\boldsymbol{x}_{t+1} = \boldsymbol{x}_t + \eta \nabla_{\boldsymbol{x}} f_{\boldsymbol{\theta}}(\boldsymbol{x})|_{\boldsymbol{x}=\boldsymbol{x}_t} , \quad \text{for } t \in [0, T-1] , \tag{3}$$

where T is the number of steps. The high-scoring design $\boldsymbol{x}^*$ can be obtained as $\boldsymbol{x}_T$. This simple gradient ascent method suffers from the out-of-distribution problem — the proxy function $f_{\boldsymbol{\theta}}(\boldsymbol{x})$ is not accurate for unseen designs, and its actual score is usually considerably lower than predicted [8].

## 2.2 Infinitely Wide DNN and Neural Tangent Kernel

Deep infinitely wide neural network models have recently received substantial attention. Previous works establish the correspondence between infinitely wide neural networks and kernel methods [11–14]. In the infinite width assumption, wide neural networks trained by gradient descent with a squared loss yield a neural tangent kernel (NTK), which is specified by the network architecture. More specifically, given two samples (designs) $\boldsymbol{x}_i$ and $\boldsymbol{x}_j$, the NTK $k(\boldsymbol{x}_i, \boldsymbol{x}_j)$ measures the similarity between the two samples from the perspective of the DNN model.

The NTKs have outperformed their finite network counterparts in various tasks and achieved state-of-the-art performance for classification tasks such as CIFAR10 [15]. The approach in [16] leverages the NTK to distill a large dataset into a small one to retain as much of its information as possible and the method in [17] uses the NTK to learn black-box generalization attacks against DNN models.

# 3 Method

In this section, we present **B**i**D**irectional learning for offline **I**nfinite-width model-based optimization (**BDI**). First, to mitigate the out-of-distribution problem, we introduce bidirectional learning between the high-scoring designs and the static dataset in Section 3.1. For a finite-width DNN model, the loss function is intractable. Approximate solutions lead to significantly poorer designs. We therefore propose the adoption of an infinite-width DNN model, and leverage its NTK to yield a closed-form loss and more accurate updates in Section 3.2.

## 3.1 Bidirectional Learning

We use $\boldsymbol{X}_h \in \mathcal{R}^{M \times D}$ to represent the high-scoring designs, where M represents the number of high-scoring designs. The high-scoring designs $\boldsymbol{X}_h$ are potentially outside of the static dataset distribution. To mitigate the out-of-distribution problem, we propose bidirectional learning between the designs and the static dataset, which leverages one to predict the other and vice versa. This can distill more information from the static dataset into the high-scoring designs. To compute the prediction losses, we need to know the scores of the high-scoring designs. Inspired by [10], in which a reward (score) is predefined for reinforcement learning, we set a predefined target score $y_h$ for every high-scoring design. More precisely, we normalize the scores of the static dataset to have unit Gaussian statistics following [8] and then choose $y_h$ to be larger than the maximal score in the static dataset. We set $y_h$ as 10 across all tasks but demonstrate that BDI is robust to different choices in Section 4.6. Thus, the high-scoring designs can be written as $(\boldsymbol{X}_h, \boldsymbol{y}_h)$, and the goal is to find $\boldsymbol{X}_h$.

**Forward mapping.** Similar to traditional gradient ascent methods, the forward mapping leverages the static dataset to predict the scores of the high-scoring designs. To be specific, the DNN model $f_{\boldsymbol{\theta}}^l(\cdot)$ trained on $(\boldsymbol{X}_l, \boldsymbol{y}_l)$ is encouraged to predict the scores $\boldsymbol{y}_h$ given $\boldsymbol{X}_h$. Thus the forward loss

function can be written as:

$$\mathcal{L}_{l2h}(\boldsymbol{X}_h) = \frac{1}{M}\|\boldsymbol{y}_h - f_{\boldsymbol{\theta}^*}^l(\boldsymbol{X}_h)\|^2 , \tag{4}$$

where $\boldsymbol{\theta}^*$ is given by

$$\boldsymbol{\theta}^* = \arg\min_{\boldsymbol{\theta}} \frac{1}{N}\|\boldsymbol{y}_l - f_{\boldsymbol{\theta}}^l(\boldsymbol{X}_l)\|^2 + \frac{\beta}{N}\|\boldsymbol{\theta}\|^2 , \tag{5}$$

where $\beta > 0$ is a fixed regularization parameter. Then we can minimize $\mathcal{L}_{l2h}(\boldsymbol{X}_h)$ against $\boldsymbol{X}_h$ to update the high-scoring designs. $\mathcal{L}_{l2h}(\boldsymbol{X}_h)$ is similar to the gradient ascent process in Eq.(3); the only difference is that gradient ascent aims to gradually increase the score prediction whereas minimizing $\mathcal{L}_{l2h}(\boldsymbol{X}_h)$ pushes the score prediction towards the (high) predefined target score $\boldsymbol{y}_h$.

**Backward mapping.** Similarly, the DNN model $f_{\boldsymbol{\theta}}^h(\cdot)$ trained on $(\boldsymbol{X}_h, \boldsymbol{y}_h)$ should be able to predict the scores $\boldsymbol{y}_l$ given $\boldsymbol{X}_l$. The backward loss function is

$$\mathcal{L}_{h2l}(\boldsymbol{X}_h) = \frac{1}{N}\|\boldsymbol{y}_l - f_{\boldsymbol{\theta}^*(\boldsymbol{X}_h)}^h(\boldsymbol{X}_l)\|^2 , \tag{6}$$

where $\boldsymbol{\theta}^*(\boldsymbol{X}_h)$ is given by

$$\boldsymbol{\theta}^*(\boldsymbol{X}_h) = \arg\min_{\boldsymbol{\theta}} \frac{1}{M}\|\boldsymbol{y}_h - f_{\boldsymbol{\theta}}^h(\boldsymbol{X}_h)\|^2 + \frac{\beta}{M}\|\boldsymbol{\theta}\|^2 . \tag{7}$$

**Overall loss.** The forward mapping and the backward mapping together align the high-scoring designs with the static dataset, and the BDI loss can be compactly written as:

$$\mathcal{L}(\boldsymbol{X}_h) = \mathcal{L}_{l2h}(\boldsymbol{X}_h) + \mathcal{L}_{h2l}(\boldsymbol{X}_h) , \tag{8}$$

where we aim to optimize the high-scoring designs $\boldsymbol{X}_h$.

## 3.2 Closed-form Solver via Neural Tangent Kernel

For a finite-width DNN model $f_{\boldsymbol{\theta}}^h(\cdot)$, the high-scoring designs $\boldsymbol{X}_h$ of Eq.(6) only exist in $\boldsymbol{\theta}^*(\boldsymbol{X}_h)$ from Eq.(7), which is intractable and does not have a closed-form solution. The solution of Eq.(7) is often approximately obtained by a gradient descent step [18, 19],

$$\boldsymbol{\theta}^*(\boldsymbol{X}_h) = \boldsymbol{\theta} - \frac{\eta}{M}\frac{\partial\|\boldsymbol{y}_h - f_{\boldsymbol{\theta}}^h(\boldsymbol{X}_h)\|^2}{\partial\boldsymbol{\theta}} , \tag{9}$$

where $\eta$ denotes the learning rate. This approximate solution $\boldsymbol{\theta}^*(\boldsymbol{X}_h)$ leads to inaccurate updates to the designs $\boldsymbol{X}_h$ through Eq.(6) and leads to much poorer designs, especially for high-dimensional settings, as we illustrate in Section 4.5.

We thus adopt a DNN model $f_{\boldsymbol{\theta}}(\cdot)$ with infinite width, and propose to leverage the corresponding NTK to produce a closed-form $\boldsymbol{\theta}^*(\boldsymbol{X}_h)$ [11, 12] and then a closed-form loss function of Eq.(6).

**Neural tangent kernel.** Denote by $k(\boldsymbol{x}_i, \boldsymbol{x}_j)$ the kernel function between $\boldsymbol{x}_i$ and $\boldsymbol{x}_j$ introduced by the infinite-width DNN $f_{\boldsymbol{\theta}}(\cdot)$. We use $\boldsymbol{K}_{\boldsymbol{X}_l\boldsymbol{X}_l} \in \mathbf{R}^{N\times N}$, $\boldsymbol{K}_{\boldsymbol{X}_h\boldsymbol{X}_h} \in \mathbf{R}^{M\times M}$, $\boldsymbol{K}_{\boldsymbol{X}_l\boldsymbol{X}_h} \in \mathbf{R}^{N\times M}$ and $\boldsymbol{K}_{\boldsymbol{X}_h\boldsymbol{X}_l} \in \mathbf{R}^{M\times N}$ to represent the corresponding covariance matrices induced by the kernel function $k(\boldsymbol{x}_i, \boldsymbol{x}_j)$. By leveraging the closed-form $\boldsymbol{\theta}^*(\boldsymbol{X}_h)$ given by [11, 12], we can compute the closed-form $\mathcal{L}_{h2l}(\boldsymbol{X}_h)$ in Eq.(6) as:

$$\mathcal{L}_{h2l}(\boldsymbol{X}_h) = \frac{1}{N}\|\boldsymbol{y}_l - \boldsymbol{K}_{\boldsymbol{X}_l\boldsymbol{X}_h}(\boldsymbol{K}_{\boldsymbol{X}_h\boldsymbol{X}_h} + \beta\boldsymbol{I})^{-1}\boldsymbol{y}_h\|^2 . \tag{10}$$

Similar to [20], to focus on higher-scoring designs in the static dataset, we assign a weight to every design based on its score and recompute the loss:

$$\mathcal{L}_{h2l}(\boldsymbol{X}_h) = \|\boldsymbol{\omega}_l \cdot (\boldsymbol{y}_l - \boldsymbol{K}_{\boldsymbol{X}_l\boldsymbol{X}_h}(\boldsymbol{K}_{\boldsymbol{X}_h\boldsymbol{X}_h} + \beta\boldsymbol{I})^{-1}\boldsymbol{y}_h)\|^2 , \tag{11}$$

where $\boldsymbol{\omega}_l = \sqrt{\text{softmax}(\alpha\boldsymbol{y}_l)}$ represents the design weight vector and the weight parameter $\alpha \geq 0$ is a constant. Similarly, we have the forward loss function,

$$\mathcal{L}_{l2h}(\boldsymbol{X}_h) = \|\boldsymbol{\omega}_h \cdot (\boldsymbol{y}_h - \boldsymbol{K}_{\boldsymbol{X}_h\boldsymbol{X}_l}(\boldsymbol{K}_{\boldsymbol{X}_l\boldsymbol{X}_l} + \beta\boldsymbol{I})^{-1}\boldsymbol{y}_l)\|^2 , \tag{12}$$

where $\boldsymbol{\omega}_h = \frac{1}{\sqrt{M}}$ since every element in $\boldsymbol{y}_h$ is the same in our paper.

Besides the advantage of the closed-form computation, introducing the NTK into the bidirectional learning can also avoid the expensive high-order derivative computations brought by Eq.(9), which makes our BDI an efficient first-order optimization method.

**Analysis of M=1.** When we aim for one high-scoring design (M = 1), Eq.(11) has a simpler form:

$$\begin{aligned}
\mathcal{L}_{h2l}(\boldsymbol{x}_h) &= \|\boldsymbol{\omega}_l \cdot (\boldsymbol{y}_l - \boldsymbol{k}_{\boldsymbol{X}_l\boldsymbol{x}_h}(k_{\boldsymbol{x}_h\boldsymbol{x}_h} + \beta)^{-1}y_h)\|^2 , \\
&= \sum_{i=1}^{N} \omega_{li}^2 (y_{li} - \frac{k(\boldsymbol{X}_{li}, \boldsymbol{x}_h)}{k_{\boldsymbol{x}_h\boldsymbol{x}_h} + \beta}y_h)^2 ,
\end{aligned} \tag{13}$$

where $k(\boldsymbol{X}_{li}, \boldsymbol{x}_h)$ measures the similarity between the $i_{th}$ design of $\boldsymbol{X}_l$ and the high-scoring design $\boldsymbol{x}_h$. Recall that $y_h$ represents the predefined target score and is generally much larger than $y_{li}$. By minimizing $\mathcal{L}_{h2l}(\boldsymbol{x}_h)$, we aim to find a high-scoring design $\boldsymbol{x}_h$ such that any design in the static dataset similar to the high scoring design is encouraged to have a high score prediction. In this way, $\boldsymbol{x}_h$ tries to incorporate as many high-scoring features from the static dataset as possible.

**Optimization.** Combining the two losses, the overall loss can be expressed as:

$$\begin{aligned}
\mathcal{L}(\boldsymbol{X}_h) = \frac{1}{2}(\|\boldsymbol{\omega}_h \cdot (\boldsymbol{y}_h - \boldsymbol{K}_{\boldsymbol{X}_h\boldsymbol{X}_l}(\boldsymbol{K}_{\boldsymbol{X}_l\boldsymbol{X}_l} + \beta\boldsymbol{I})^{-1}\boldsymbol{y}_l)\|^2 \\
+ \|\boldsymbol{\omega}_l \cdot (\boldsymbol{y}_l - \boldsymbol{K}_{\boldsymbol{X}_l\boldsymbol{X}_h}(\boldsymbol{K}_{\boldsymbol{X}_h\boldsymbol{X}_h} + \beta\boldsymbol{I})^{-1}\boldsymbol{y}_h)\|^2) ,
\end{aligned} \tag{14}$$

where only the high-scoring designs $\boldsymbol{X}_h$ are learnable parameters in the whole procedure of BDI. We optimize $\boldsymbol{X}_h$ against $\mathcal{L}(\boldsymbol{X}_h)$ by gradient descent methods such as Adam [21].

# 4 Experiments

We conduct extensive experiments on design-bench [1], and aim to answer three research questions: (1) How does BDI compare with both recently proposed offline model-based algorithms and traditional algorithms? (2) Is every component necessary in BDI? (i.e., $\mathcal{L}_{h2l}(\boldsymbol{X}_h)$, $\mathcal{L}_{l2h}(\boldsymbol{X}_h)$, NTK.) (3) Is BDI robust to the hyperparameter choices including the predefined target score $y_h$, the weight parameter $\alpha$ and the number of steps T?

## 4.1 Dataset and Evaluation

To evaluate the effectiveness of BDI, we adopt the commonly used design-bench, including both continuous and discrete tasks, and the evaluation protocol as in the prior work [8].

**Task overview.** We conduct experiments on four continuous tasks: **(a)** Superconductor (SuperC) [2], where the aim is to design a superconductor with 86 continuous components to maximize critical temperature with 17010 designs; **(b)** Ant Morphology (Ant) [1, 22], where the goal is to design the morphology of a quadrupedal ant with 60 continuous components to crawl quickly with 10004 designs, **(c)** D'Kitty Morphology (D'Kitty) [1, 23], where the goal is to design the morphology of a quadrupedal D'Kitty with 56 continuous components to crawl quickly with 10004 designs; and **(d)** Hopper Controller (Hopper) [1], where the aim is to find a neural network policy parameterized by 5126 total weights to maximize return with 3200 designs. Besides the continuous tasks, we perform experiments on three discrete tasks: **(e)** GFP [3], where the objective is to find a length 238 protein sequence to maximize fluorescence with 5000 designs; **(f)** TF Bind 8 (TFB) [5], where the aim is to find a length 8 DNA sequence to maximize binding activity score with 32896 designs; and **(g)** UTR [6], where the goal is to find a length 50 DNA sequence to maximize expression level with $140,000$ designs. These tasks contain neither personally identifiable nor offensive information.

**Evaluation.** We follow the same evaluation protocol in [8]: we choose the top N = 128 most promising designs for each method, and then report the $100^{th}$ percentile normalized ground truth score as $y_n = \frac{y - y_{min}}{y_{max} - y_{min}}$ where $y_{min}$ and $y_{max}$ represent the lowest score and the highest score in the full unobserved dataset, respectively. The additional $50^{th}$ percentile (median) normalized ground truth scores, used in the prior work [8], are also provided in Appendix A.1. To better measure the performance across multiple tasks, we further report the mean and median ranks of the compared algorithms over all 7 tasks.

## 4.2 Comparison Methods

We compare BDI with two groups of baselines: (i) sampling via a generative model; and (ii) gradient updating from existing designs. The generative model based methods learn a distribution of the high-scoring designs and then sample from the learned distribution. This group of methods includes: 1) CbAS [24]: trains a VAE model of the design distribution using designs with a score above a threshold and gradually adapts the distribution to the high-scoring part by increasing the threshold. 2) Auto.CbAS [25]: uses importance sampling to retrain a regression model using the design distribution introduced by CbAS; 3) MIN [20]: learns an inverse map from the score to the design and queries the inverse map by searching for the optimal score to obtain the optimal design. The latter group includes: 1) Grad: applies simple gradient ascent on existing designs to obtain new designs; 2) ROMA [7]: regularizes the smoothness of the DNN and then performs gradient ascent to obtain new designs; 3) COMs [8]: regularizes the DNN to assign lower scores to designs obtained during the gradient ascent process and leverages this model to obtain new designs by gradient ascent; 4) NEMO [9]: bounds the distance between the proxy and the objective function by normalized maximum likelihood and then performs gradient ascent. We discuss the NTK based Grad in Section 4.5.

Besides the recent work, we also compare BDI with several traditional methods described in [1]: 1) BO-qEI [26]: performs Bayesian Optimization to maximize the proxy, proposes designs via the quasi-Expected-Improvement acquisition function, and labels the designs via the proxy function; 2) CMA-ES [27]: an evolutionary algorithm gradually adapts the distribution via the covariance matrix towards the optimal design; 3) REINFORCE [28]: first learns a proxy function and then optimizes the distribution over the input space by leveraging the proxy and the policy-gradient estimator.

## 4.3 Training Details

We follow the training settings of [8] for all comparison methods if not specified. Recall that $M$ represents the number of the high-scoring designs. We set $M = 1$ in our experiments, which achieves expressive results, and discuss the $M > 1$ scenario in Appendix A.2. We set the regularization $\beta$ as $1e^{-6}$ following [16]. We set the predefined target score $y_h$ as a constant 10 across all tasks, which proves to be effective and robust in all cases. We set $\alpha$ as $1e^{-3}$ for all continuous tasks and as $0.0$ for all discrete tasks, and set the number of iterations $T$ to 200 in all experiments. We discuss the choices of $y_h$, $\alpha$ and $T$ in Sec. 4.6 in detail. We adopt a 6-layer MLP (MultiLayer Perceptron) followed by ReLU for all gradient updating methods and the hidden size is set as 2048. We optimize $\mathcal{L}(\boldsymbol{X}_h)$ with the Adam optimizer [21] with a $1e^{-1}$ learning rate for discrete tasks and a $1e^{-3}$ learning rate for continuous tasks. We cite the results from [8] for the non-gradient-ascent methods including BO-qEI, CMA-ES[2], REINFORCE, CbAS, Auto.CbAS. For other methods, we run every setting over 8 trials and report the mean and the standard error. We use the NTK library [29] build on Jax [30] to conduct kernel-based experiments and use Pytorch [31] for other experiments. All Jax experiments are run on multiple CPUs within a cluster and all Pytorch experiments are run on one V100 GPU. We discuss the computational time details for all tasks in Appendix A.3.

## 4.4 Results and Analysis

We report experimental results in Table 1 for continuous tasks and in Table 2 for discrete tasks where $\mathcal{D}(\textbf{best})$ represents the maximal score of the static dataset for each task. We bold the results within one standard deviation of the highest performance.

**Results on continuous tasks.** As shown in Table 1, BDI achieves the best results over all tasks. Compared with the naive Grad method, BDI achieves consistent gain over all four tasks, which suggests that BDI can align the high-scoring design with the static dataset and thus mitigate the out-of-distribution problem. As for COMs, ROMA and NEMO, which all impose a prior on the DNN model, they generally perform better than Grad but worse than BDI. This indicates that directly transforming the information of the static dataset into the high-scoring design is more effective than adding priors on the DNN in terms of mitigating the out-of-distribution problem. Another advantage of BDI is in its neural tangent kernel nature: 1) Real-world offline model-based optimization often involves small-scale datasets since the labeling cost of protein/DNA/robot is very high; 2) BDI is built on the neural tangent kernel, which generally performs better than finite neural networks on

---

[2]CMA-ES performs very differently on Ant Morphology and D'Kitty Morphology and the reason is that Ant Morphology is simpler and much more sensitive to initializations.

Table 1: Experimental results on continuous tasks for comparison.

| Method | Superconductor | Ant Morphology | D'Kitty Morphology | Hopper Controller |
|---|---|---|---|---|
| $\mathcal{D}(\textbf{best})$ | 0.399 | 0.565 | 0.884 | 1.0 |
| BO-qEI | $0.402 \pm 0.034$ | $0.819 \pm 0.000$ | $0.896 \pm 0.000$ | $0.550 \pm 0.118$ |
| CMA-ES | $0.465 \pm 0.024$ | $\textbf{1.214} \pm \textbf{0.732}$ | $0.724 \pm 0.001$ | $0.604 \pm 0.215$ |
| REINFORCE | $0.481 \pm 0.013$ | $0.266 \pm 0.032$ | $0.562 \pm 0.196$ | $-0.020 \pm 0.067$ |
| CbAS | $0.503 \pm 0.069$ | $0.876 \pm 0.031$ | $0.892 \pm 0.008$ | $0.141 \pm 0.012$ |
| Auto.CbAS | $0.421 \pm 0.045$ | $0.882 \pm 0.045$ | $0.906 \pm 0.006$ | $0.137 \pm 0.005$ |
| MIN | $0.452 \pm 0.026$ | $0.908 \pm 0.020$ | $\textbf{0.942} \pm \textbf{0.010}$ | $0.094 \pm 0.113$ |
| Grad | $0.483 \pm 0.027$ | $0.764 \pm 0.047$ | $0.888 \pm 0.035$ | $0.989 \pm 0.362$ |
| COMs | $0.487 \pm 0.023$ | $0.866 \pm 0.050$ | $0.835 \pm 0.039$ | $1.224 \pm 0.555$ |
| ROMA | $0.476 \pm 0.024$ | $0.814 \pm 0.051$ | $0.905 \pm 0.018$ | $1.849 \pm 0.110$ |
| NEMO | $0.488 \pm 0.034$ | $0.814 \pm 0.043$ | $0.924 \pm 0.012$ | $\textbf{1.959} \pm \textbf{0.159}$ |
| $\textbf{BDI}_{\text{(ours)}}$ | $\textbf{0.520} \pm \textbf{0.005}$ | $\textbf{0.962} \pm \textbf{0.000}$ | $\textbf{0.941} \pm \textbf{0.000}$ | $\textbf{1.989} \pm \textbf{0.050}$ |

Table 2: Experimental results on discrete tasks & ranking on all tasks for comparison.

| Method | GFP | TF Bind 8 | UTR | Rank Mean | Rank Median |
|---|---|---|---|---|---|
| $\mathcal{D}(\textbf{best})$ | 0.789 | 0.439 | 0.593 | | |
| BO-qEI | $0.254 \pm 0.352$ | $0.798 \pm 0.083$ | $0.684 \pm 0.000$ | 8.6/11 | 9/11 |
| CMA-ES | $0.054 \pm 0.002$ | $0.953 \pm 0.022$ | $0.707 \pm 0.014$ | 5.7/11 | 6/11 |
| REINFORCE | $\textbf{0.865} \pm \textbf{0.000}$ | $0.948 \pm 0.028$ | $0.688 \pm 0.010$ | 7.4/11 | 9/11 |
| CbAS | $\textbf{0.865} \pm \textbf{0.000}$ | $0.927 \pm 0.051$ | $0.694 \pm 0.010$ | 4.6/11 | 5/11 |
| Auto.CbAS | $\textbf{0.865} \pm \textbf{0.000}$ | $0.910 \pm 0.044$ | $0.691 \pm 0.012$ | 6.0/11 | 7/11 |
| MIN | $\textbf{0.865} \pm \textbf{0.001}$ | $0.882 \pm 0.020$ | $0.694 \pm 0.017$ | 5.3/11 | 4/11 |
| Grad | $0.864 \pm 0.001$ | $0.491 \pm 0.042$ | $0.666 \pm 0.013$ | 7.9/11 | 8/11 |
| COMs | $0.861 \pm 0.009$ | $0.920 \pm 0.043$ | $0.699 \pm 0.011$ | 5.6/11 | 6/11 |
| ROMA | $0.558 \pm 0.395$ | $0.928 \pm 0.038$ | $0.690 \pm 0.012$ | 6.1/11 | 7/11 |
| NEMO | $0.150 \pm 0.270$ | $0.905 \pm 0.048$ | $0.694 \pm 0.015$ | 5.4/11 | 4/11 |
| $\textbf{BDI}_{\text{(ours)}}$ | $0.864 \pm 0.000$ | $\textbf{0.973} \pm \textbf{0.000}$ | $\textbf{0.760} \pm \textbf{0.000}$ | $\textbf{1.9/11}$ | $\textbf{1/11}$ |

small-scale datasets [32]; 3) COMs, ROMA, and NEMO are built on a finite neural network and it is non-trivial to modify them to the neural tangent kernel version. The generative model based methods CbAS, Auto.CbAS and MIN perform poorly for high-dimensional tasks like Hopper (D=5126) since the high-dimensional data distributions are harder to model. Compared with generative model based methods, BDI not only achieves better results but is also much simpler.

**Results on discrete tasks.** As shown in Table 2, BDI achieves the best or equal-best performances in two of the three tasks, and is only marginally inferior in the third. This suggests that BDI is also a powerful approach in the discrete domain. For the GFP task, every design is a length 238 sequence of 20-categorical one-hot vectors and gradient updates that do not take into account the sequential nature may be less beneficial. This may explain why BDI does not perform as well on GFP.

**Overall,** BDI achieves the best result in terms of the ranking as shown in Table 2 and Figure 2, and attains the best performance on **6/7** tasks.

Table 3: Ablation studies on BDI components.

| Task | D | BDI | w/o $\mathcal{L}_{l2h}$ | w/o $\mathcal{L}_{h2l}$ | NTK2NN | NTK2RBF |
|---|---|---|---|---|---|---|
| GFP | 238 | $\textbf{0.864} \pm \textbf{0.000}$ | $0.860 \pm 0.004$ | $\textbf{0.864} \pm \textbf{0.000}$ | $0.252 \pm 0.354$ | $0.862 \pm 0.003$ |
| TFB | 8 | $0.973 \pm 0.000$ | $\textbf{0.984} \pm \textbf{0.000}$ | $0.973 \pm 0.000$ | $0.958 \pm 0.025$ | $0.925 \pm 0.000$ |
| UTR | 50 | $0.760 \pm 0.000$ | $0.636 \pm 0.000$ | $\textbf{0.777} \pm \textbf{0.000}$ | $0.699 \pm 0.009$ | $0.738 \pm 0.000$ |
| SuperC | 86 | $\textbf{0.520} \pm \textbf{0.005}$ | $0.511 \pm 0.007$ | $0.502 \pm 0.007$ | $0.450 \pm 0.036$ | $0.510 \pm 0.012$ |
| Ant | 60 | $\textbf{0.962} \pm \textbf{0.000}$ | $0.914 \pm 0.000$ | $0.933 \pm 0.000$ | $0.861 \pm 0.019$ | $0.927 \pm 0.000$ |
| D'Kitty | 56 | $0.941 \pm 0.000$ | $0.920 \pm 0.000$ | $\textbf{0.942} \pm \textbf{0.001}$ | $0.929 \pm 0.010$ | $0.938 \pm 0.008$ |
| Hopper | 5126 | $\textbf{1.989} \pm \textbf{0.050}$ | $1.935 \pm 0.553$ | $1.821 \pm 0.223$ | $0.503 \pm 0.039$ | $0.965 \pm 0.103$ |

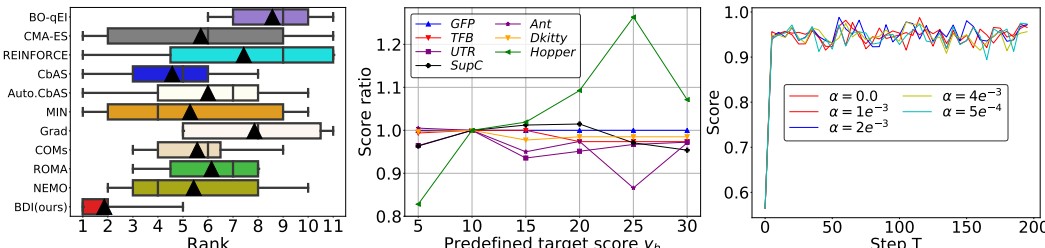

Figure 2: Rank minima and maxima are indicated by whiskers; vertical lines and black triangles represent medians and means.

Figure 3: **The ratio of** the ground truth score of the design with $y_h$ **to** the ground truth score with $y_h = 10$.

Figure 4: The ground truth score of the design as a function of the step T for different weight parameters $\alpha$ on Ant.

## 4.5 Ablation Studies

In this subsection, we conduct ablation studies on the components of BDI and aim to verify the effectiveness of $\mathcal{L}_{h2l}(\boldsymbol{X}_h)$, $\mathcal{L}_{l2h}(\boldsymbol{X}_h)$, and NTK. The baseline method is our proposed BDI and we remove each component to verify its effectiveness. We replace the NTK with its corresponding finite DNN (6-layer MLP followed by ReLU), which is trained on $(\boldsymbol{X}_l, \boldsymbol{y}_l)$ for Eq.(5) and $(\boldsymbol{X}_h, \boldsymbol{y}_h)$ for Eq.(7). Following [16], another alternative is to replace the NTK with RBF.

As shown in Table 3, BDI generally performs the best (at least second-best) across tasks, which indicates the importance of all three components.

**Forward mapping.** Removing $\mathcal{L}_{l2h}(\boldsymbol{X}_h)$ yields the best result $0.984$ in the TFB task, but worsens the designs in other cases, which verifies the importance of the forward mapping. Incorporating the $\mathcal{L}_{l2h}(\boldsymbol{X}_h)$ loss term is similar to using the Grad method as we discuss in Section 3.1. We also run experiments on gradient ascent with NTK. This approach first builds a regression model using the NTK and then performs gradient ascent on existing designs to obtain new designs. Results are very similar to BDI without $\mathcal{L}_{h2l}(\boldsymbol{X}_h)$, so we do not report them here.

**Backward mapping.** Removing $\mathcal{L}_{h2l}(\boldsymbol{X}_h)$ decreases model performances in three tasks, and only improves it for the UTR task. This demonstrates the importance of the backward mapping. Although removing $\mathcal{L}_{h2l}$ leads to the best result $0.777$ in UTR compared with $0.760$ of BDI, this does not necessarily demonstrate that the removal of $\mathcal{L}_{h2l}$ is desirable. The $50^{th}$ percentile of BDI is higher than that of BDI without $\mathcal{L}_{h2l}$ ($0.664 > 0.649$). Furthermore, we observe that the backward mapping is more effective for continuous tasks, possibly because we do not model the sequential nature of DNA and protein sequences. We follow the default procedure in [8] which maps the sequence design to real-valued logits of a categorical distribution for a fair comparison. Given a more effective discrete space modeling, the backward mapping might yield better results; we leave this to future work.

**Neural tangent kernel.** We replace the NTK with the corresponding finite DNN (NTK2NN) and the RBF kernel (NTK2RBF), and both replacements degrade the performance significantly, which verifies the effectiveness of the NTK. In addition, we can observe that NTK2NN performs relatively well on the low-dimensional task TFB (D=8) but poorly on high-dimensional tasks, especially for GFP (D = 238) and Hopper (D = 5126). It is likely that the high-dimensional designs $\boldsymbol{X}_h$ are more sensitive to the approximation in Eq. (9). Furthermore, BDI performs better than NTK2RBF and this can be explained as follows. Compared with the RBF kernel, the NTK enjoys the high expressiveness of a DNN [15] and represents a broad class of DNNs [16, 17], which can better extract the design features and enhance the generalization of the high-scoring designs.

## 4.6 Hyperparameter Sensitivity

In this subsection, we first study the sensitivity of BDI to the predefined target score $y_h$. Note that all algorithms in this subsection do not have access to the ground truth scores, which are only used for analysis. We expect this score to be larger than the maximal score of the static dataset since we want to obtain designs better than those in the static dataset. We first normalize all scores to have unit Gaussian statistics following [1, 8]. A small target score may not be able to identify a high-scoring design and a large score is too hard to reach. Thus we choose to set the target score $y_h$ as 10 across all

Table 4: Comparison between data distillation and backward mapping.

| Comparison | Data distillation | Backward mapping |
|---|---|---|
| Data $x$ | images in the training dataset $\mathcal{D}$ | designs in the offline dataset $\mathcal{D}$ |
| Label content | 0-9 for the MNIST dataset | some measurement of protein/dna/robot |
| Label value | within the training dataset $\mathcal{D}$ | larger than the max of the offline dataset $\mathcal{D}$ |
| Loss function | cross-entropy for classification | mean squared error for regression |
| Task objective | distill $\mathcal{D}$ into a small subset | distill $x$ to incorporate high-scoring features |

tasks. In Figure 3, we report **the ratio of** the ground truth score of the design with $y_h$ **to** the ground truth score of the design with $y_h = 10$ and the score details are in Appendix A.4. All tasks are robust to the change of $y_h$. Although we set $y_h$ as 10, which is smaller than the maximal score 10.2 for Hopper, BDI still yields a good result, which demonstrates the robustness of BDI. For Hopper, after we set $y_h$ to a larger value, we can observe that the corresponding results become better.

Besides $y_h$, we also explore robustness to the choice of the weight parameter $\alpha$ and the number of steps T. We study the continuous task Ant and the discrete task TFB. As shown in Figure 4, we visualize the $100^{th}$ percentile ground truth score of the Ant task as a function of T for $\alpha = 0.0, 5e^{-4}, 1e^{-3}, 2e^{-3}, 4e^{-3}$. We can observe that the behavior of BDI is robust to change of $\alpha$ within the range of evaluated values. BDI yields the high-scoring designs at early steps and then the solutions remain stable. This demonstrates the robustness of BDI to the choice of T. For the TFB task, we find the behavior of BDI is identical for different $\alpha$ and the behavior with respect to T is very similar to that of Ant. Thus we do not report details here. One possible reason for the insensitivity to $\alpha$ is that TFB is a discrete task and the discrete solution is more robust to $\alpha$.

## 5   Related Work

**Offline model-based optimization.** Recent methods for offline model-based optimization can be divided into two categories: (i) obtaining designs from a generative model and (ii) applying gradient ascent on existing designs. In the first category, the methods in [24, 25] gradually adapt a VAE-based generative model towards the optimized design by leveraging a proxy function. Kumar et al. adopt an alternative approach in [20], learning a generator that maps the score to the design. The generative model based methods often require careful tuning to model the high-scoring designs [8].

Gradient-based methods have received substantial attention recently because they can leverage DNN models to produce improved designs. A challenge in the naive application of such methods is out-of-distribution designs, for which the trained DNN model produces inaccurate score predictions. Several approaches have been proposed to address this. Trabucco et al. add a regularizing term for conservative objective modeling [8, 33]. In [9], Fu et al. leverage normalized maximum likelihood to bound the distance between the proxy function and the ground truth. Yu et al. overcome the brittleness of the proxy function by utilizing a local smoothness prior [7]. Our work falls into the gradient-based line of research, but rather than imposing a prior on the *model*, we propose the bidirectional mapping to ensure that the proposed designs can be used to predict the scores of the *data* and vice versa, thus ensuring that the high-scoring designs are more aligned with the static dataset. Similar to our backward mapping constraining designs, BCQ [34] constrains the state-action pairs to be contained in the support of the static dataset. Our proposed backward mapping is different in its implementation and also enables the model to generate designs outside the training distribution.

**Data distillation.** Data distillation [18, 35–38] aims to distill a large training set into a small one, which enables efficient retraining. Building upon [18], the methods in [39, 40] propose to distill labels instead of images. Approaches in [16, 41] introduce a meta-learning algorithm powered by NTK, which significantly improves the performance. The backward mapping of BDI is inspired by [16, 41] and the difference is that BDI is handling a regression problem with a predefined target score larger than the maximal score of the static dataset. In Table 4 we provided a detailed comparison of backward mapping and data distillation; both adopt a bi-level framework [42]. This bi-level framework can also be used to learn other hyperaparameters [17, 43–47].

# 6 Conclusion and discussion

In this paper, we address offline model-based optimization and propose a novel approach that involves bidirectional score predictions. BDI requires the proposed high-scoring designs and the static offline dataset to predict each other, which effectively ameliorates the out-of-distribution problem. The finite-width DNN can only yield an approximate loss function, which leads to a significant deterioration of the design quality, especially for high-dimensional tasks. We thus adopt an infinite-width DNN and employ the NTK to yield a closed-form solution, leading to better designs. Experimental results verify the effectiveness of BDI, demonstrating that it outperforms state-of-the-art approaches.

**Limitations.** Although BDI has been demonstrated to be effective in tasks from a wide range of domains, some evaluations are not performed with a realistic setup. For example, in some of our experiments, such as the GFP [3] task, we follow the prior work and use a transformer model which is pre-trained on a large dataset as the evaluation oracle. However, this may not reflect real-world behavior and the evaluation protocol can be further improved by close collaboration with domain experts [48–50]. Overall, we still believe BDI can generalize well to these cases since BDI has a simple formulation and proves to be effective and robust for the wide range of tasks in the design-bench.

**Negative impacts.** Designing a new object or entity with desired properties is a double-edged sword. On one hand, it can be beneficial to society, with an example being drug discovery to cure unsolved diseases. On the other hand, if these techniques are acquired by dangerous people, they can also be used to design biochemicals to harm our society. Researchers should pay vigilant attention to ensure their research does end up being used positively for the social good.

# 7 Acknowledgement

We thank Aristide Baratin from Mila and Greg Yang from Microsoft Research for their helpful discussions on the neural tangent kernel and thank Zixuan Liu from the University of Washington for his helpful suggestions on the paper writing. This research was supported in part by funding from the Fonds de recherche du Québec – Nature et technologies.

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
