# A Appendix

## A.1 Additional Results on 50th Percentile Scores

In the main paper, we have provided results for the $100^{th}$ percentile metric and here we provide additional results on the $50^{th}$ percentile metric, also used in the prior work design-bench, to verify the effectiveness of BDI. Table 5 and Table 6, report $50^{th}$ percentile results for continuous tasks and discrete tasks, respectively. We can observe from Table 6 that BDI achieves the best result in terms of ranking. Note that there is no randomness for BDI by the nature of the method, once the initial design has been selected. To quantify the robustness of BDI to this starting design, we perturb each initial design by performing a small gradient ascent step according to a trained DNN model on the design. The mean and standard deviation are obtained over 8 different seeds. We can see that BDI is robust to the randomness.

## A.2 Analysis of multiple high-scoring designs

We now discuss the high-scoring design number M >1 scenarios and consider two cases: (a) all M high-scoring designs are learnable; (b) only one high-scoring design is learnable and the other $M - 1$ designs are fixed. In both cases, we set the predefined target score $y_h$ as 10. For case (a), we first choose a design from the static dataset and perform $M-1$ gradient ascent steps on the design to obtain the $M-1$ designs. Then BDI is performed between the static dataset and the M designs. Finally, the design with the maximal predicted score among the M designs is chosen for evaluation. For case (b), we choose $M-1$ designs from the static dataset along with their ground-truth scores and then perform our BDI algorithm.

We report the results of cases (a) and (b) in Table 7 and Table 8, respectively. In both cases the M >1 settings work well but are generally worse than M = 1. One possible explanation is that BDI cannot distill sufficient information from the static dataset into the high-scoring design when the other $M - 1$ designs also contribute to the backward mapping. As a result, the M = 1 setting yields a better result.

## A.3 Computational Complexity

The only learnable part of BDI is $\boldsymbol{X}_h$. Other coefficients, like $(\boldsymbol{K}_{\boldsymbol{X}_l\boldsymbol{X}_l} + \beta\boldsymbol{I})^{-1}\boldsymbol{y}_l$ in Eq. (12), can be pre-computed. Thus the computational time of BDI consists of two parts: pre-computation and gradient update. The main bottleneck of pre-computation lies in $(\boldsymbol{K}_{\boldsymbol{X}_l\boldsymbol{X}_l} + \beta\boldsymbol{I})^{-1}\boldsymbol{y}_l$, which results in the $\mathcal{O}(\mathrm{N}^3)$ time complexity, where N is the size of the static dataset $\mathcal{D}$. Note that the precomputation time includes the kernel matrix construction. The N of the static dataset usually is not large which means the computational time is usually reasonable, especially considering the potentially enormous cost of evaluating the unknown objective function later in the design cycle.

In many real-world settings, for example, those that involve biological or chemical experiments, the majority of the time in the production cycle will be spent on evaluating the unknown objective function. As a result, the time difference between the methods for obtaining the high score design is less significant in a real production environment.

Table 5: Experimental results of $50^{th}$ on continuous tasks for comparison.

| Method | Superconductor | Ant Morphology | D'Kitty Morphology | Hopper Controller |
|---|---|---|---|---|
| $\mathcal{D}(\textbf{best})$ | 0.399 | 0.565 | 0.884 | 1.000 |
| BO-qEI | $0.300 \pm 0.015$ | $0.567 \pm 0.000$ | $\textbf{0.883} \pm \textbf{0.000}$ | $0.343 \pm 0.010$ |
| CMA-ES | $0.379 \pm 0.003$ | $-0.045 \pm 0.004$ | $0.684 \pm 0.016$ | $-0.033 \pm 0.005$ |
| REINFORCE | $\textbf{0.463} \pm \textbf{0.016}$ | $0.138 \pm 0.032$ | $0.356 \pm 0.131$ | $-0.064 \pm 0.003$ |
| CbAS | $0.111 \pm 0.017$ | $0.384 \pm 0.016$ | $0.753 \pm 0.008$ | $0.015 \pm 0.002$ |
| Auto.CbAS | $0.131 \pm 0.010$ | $0.364 \pm 0.014$ | $0.736 \pm 0.025$ | $0.019 \pm 0.008$ |
| MIN | $0.325 \pm 0.021$ | $\textbf{0.606} \pm \textbf{0.025}$ | $\textbf{0.886} \pm \textbf{0.003}$ | $0.027 \pm 0.115$ |
| Grad | $0.400 \pm 0.059$ | $0.311 \pm 0.046$ | $0.736 \pm 0.168$ | $0.195 \pm 0.086$ |
| COMs | $0.365 \pm 0.045$ | $0.386 \pm 0.083$ | $0.685 \pm 0.022$ | $0.322 \pm 0.032$ |
| ROMA | $0.354 \pm 0.014$ | $0.383 \pm 0.029$ | $0.775 \pm 0.009$ | $0.358 \pm 0.011$ |
| NEMO | $0.359 \pm 0.021$ | $0.390 \pm 0.028$ | $0.800 \pm 0.020$ | $0.345 \pm 0.034$ |
| $\textbf{BDI}_{(\text{ours})}$ | $0.408 \pm 0.012$ | $0.569 \pm 0.000$ | $0.875 \pm 0.001$ | $\textbf{0.419} \pm \textbf{0.003}$ |

As for the gradient update, its computation bottleneck lies in the forward and backward passes of NTK computation. We report the details of BDI computational time on different tasks in Table 9. All experiments are performed on CPUs in the same cluster. For other gradient-based methods, the computational time is on the same scale as BDI.

## A.4 Predefined Target Scores

We report the unnormalized experimental results on different predefined target scores in Table 10 and these results correspond to Figure 3 in the main paper. Recall that the ground truth score returned by the evaluation oracle is $y$ and we compute the normalized ground truth score as $y_n = \frac{y - y_{min}}{y_{max} - y_{min}}$ where $y_{min}$ and $y_{max}$ represent the lowest score and the highest score in the full unobserved dataset, respectively. We also report the normalized experimental results in Table 11. We can observe all tasks are robust to $y_h$. Although we set $y_h$ as 10, which is smaller than the maximal score 10.2 for Hopper, BDI still yields a good result, which demonstrates the robustness of BDI.

## A.5 Considering Sequential Nature

We adopt DNABert [51] and ProtBert [52] to model the sequential nature of DNA/protein sequences. We conduct an ablation study, removing the backward mapping to verify its effectiveness. We also remove the forward mapping to assess its importance. The following experiments demonstrate our intuition: **backward mapping matters**.

For the DNA experiments, we consider the task with an exact oracle: TFBind8, and obtain the 100th percentile results in Table 12. We can see that both the backward mapping and the forward mapping matter in BDI. We further consider a harder task setting TFBind8(reduce) or TFBind8(r): reduce the size of the offline dataset from 32896 to 5000. The results verify the effectiveness and the robustness of bidirectional mappings.

For the protein experiments, we first consider the GFP task we used in our paper, and obtain the same 0.864 results for all variants, which is also mentioned in [1] "lack in-distinguishable results across all methods". To further investigate this problem, we introduce 5 new protein tasks used in recent works: LACT, AMP, ALIP, LEV, and SUMO, and present the task details and the results in the following.

- LACT: We measure the thermodynamic stability of the TEM-1 $\beta$-Lactamase protein sequence. The oracle is trained on 17857 samples from [53, 54]. The bottom half of these samples are used for offline algorithms.

- AMP: Antimicrobial peptides are short protein sequences that act against pathogens. We use the 6760 AMPs from [55] to train the oracle and the bottom half of the AMPs for the offline algorithms.

- ALIP: We measure the enzyme activity of the Aliphatic Amide Hydrolase sequence following [56]. We use the 6629 samples from [56] to train the oracle and the bottom half of them for the offline algorithms.

Table 6: Experimental results of $50^{th}$ on discrete tasks & ranking on all tasks for comparison.

| Method | GFP | TF Bind 8 | UTR | Rank Mean | Rank Median |
|---|---|---|---|---|---|
| $\mathcal{D}(\textbf{best})$ | 0.789 | 0.439 | 0.593 | | |
| BO-qEI | $0.246 \pm 0.341$ | $0.439 \pm 0.000$ | $0.571 \pm 0.000$ | 6.0/11 | 5/11 |
| CMA-ES | $0.047 \pm 0.000$ | $0.537 \pm 0.014$ | $0.612 \pm 0.014$ | 7.1/11 | 10/11 |
| REINFORCE | $0.844 \pm 0.003$ | $0.462 \pm 0.021$ | $0.568 \pm 0.017$ | 7.4/11 | 10/11 |
| CbAS | $\textbf{0.852} \pm \textbf{0.004}$ | $0.428 \pm 0.010$ | $0.572 \pm 0.023$ | 7.4/11 | 9/11 |
| Auto.CbAS | $0.848 \pm 0.007$ | $0.419 \pm 0.007$ | $0.576 \pm 0.011$ | 7.9/11 | 8/11 |
| MIN | $0.427 \pm 0.012$ | $0.421 \pm 0.015$ | $0.586 \pm 0.007$ | 6.0/11 | 7/11 |
| Grad | $0.836 \pm 0.000$ | $0.439 \pm 0.000$ | $0.611 \pm 0.000$ | 5.4/11 | 5/11 |
| COMs | $0.737 \pm 0.262$ | $0.512 \pm 0.051$ | $0.608 \pm 0.012$ | 5.3/11 | 5/11 |
| ROMA | $0.541 \pm 0.382$ | $0.439 \pm 0.000$ | $0.592 \pm 0.005$ | 5.4/11 | 5/11 |
| NEMO | $0.146 \pm 0.260$ | $0.439 \pm 0.000$ | $0.590 \pm 0.007$ | 5.4/11 | 5/11 |
| $\textbf{BDI}_{(\text{ours})}$ | $\textbf{0.860} \pm \textbf{0.010}$ | $\textbf{0.595} \pm \textbf{0.000}$ | $\textbf{0.664} \pm \textbf{0.000}$ | **1.6/11** | **1/11** |

Table 7: Experimental results on different Ms with all high-scoring designs learnable.

| Task | $\mathcal{D}_{\textbf{best}}$ | M = 1 | M = 2 | M = 3 | M = 4 |
|------|------|------|------|------|------|
| GFP | 0.789 | **0.864** | **0.864** | **0.864** | **0.864** |
| TFB | 0.439 | **0.973** | **0.973** | **0.973** | **0.973** |
| UTR | 0.593 | 0.760 | **0.762** | 0.757 | 0.757 |
| SuperC | 0.399 | **0.520** | 0.502 | 0.485 | 0.510 |
| Ant | 0.565 | **0.962** | 0.323 | 0.275 | 0.277 |
| D'Kitty | 0.884 | **0.941** | 0.717 | 0.718 | 0.718 |
| Hopper | 1.000 | **1.989** | 1.412 | 1.299 | 1.319 |

Table 8: Experimental results on different Ms with one high-scoring design learnable.

| Task | $\mathcal{D}_{\textbf{best}}$ | M = 1 | M = 2 | M = 3 | M = 4 |
|------|------|------|------|------|------|
| GFP | 0.789 | **0.864** | 0.854 | 0.857 | 0.858 |
| TFB | 0.439 | **0.973** | 0.779 | 0.934 | 0.934 |
| UTR | 0.593 | **0.760** | 0.728 | 0.720 | 0.720 |
| SuperC | 0.399 | **0.520** | 0.508 | 0.483 | 0.485 |
| Ant | 0.565 | **0.962** | 0.169 | 0.174 | 0.214 |
| D'Kitty | 0.884 | **0.941** | 0.721 | 0.720 | 0.719 |
| Hopper | 1.000 | **1.989** | 1.378 | 1.815 | 1.354 |

Table 9: Computational time of BDI on different tasks.

| Task | GFP | TFB | UTR | SuperC | Ant | D'Kitty | Hopper |
|------|------|------|------|------|------|------|------|
| Size N | 5000 | 32896 | 140000 | 17010 | 10004 | 10004 | 3200 |
| Pre(mins) | 0.2 | 4.1 | 64.5 | 1.2 | 0.5 | 0.5 | 0.1 |
| Grad(mins) | 67.1 | 9.3 | 93.6 | 9.1 | 7.6 | 9.8 | 54.6 |
| Total(mins) | 67.3 | 13.4 | 158.1 | 10.3 | 8.1 | 10.3 | 54.7 |

Table 10: Experimental results on different predefined target scores.

| Task | $\mathcal{D}_{\textbf{best}}$ | 5 | 10 | 15 | 20 | 25 | 30 |
|------|------|------|------|------|------|------|------|
| GFP | 1.444 | **6.294** | **6.294** | **6.294** | **6.294** | **6.294** | **6.294** |
| TFB | 1.480 | 8.409 | **8.462** | **8.462** | 8.239 | 8.239 | 8.239 |
| UTR | 7.116 | 8.796 | **9.120** | 8.532 | 8.676 | 8.820 | 8.868 |
| SuperC | 2.652 | 3.609 | 3.746 | 3.792 | **3.801** | 3.637 | 3.573 |
| Ant | 1.756 | **5.554** | 5.526 | 5.250 | 5.383 | 4.785 | 5.364 |
| D'Kitty | 0.934 | 1.190 | **1.194** | 1.167 | 1.176 | 1.176 | 1.176 |
| Hopper | 10.221 | 19.797 | 23.906 | 24.363 | 26.121 | **30.203** | 25.609 |

Table 11: Experimental results on different predefined target scores (normalized).

| Task | $\mathcal{D}_{\textbf{best}}$ | 5 | 10 | 15 | 20 | 25 | 30 |
|------|------|------|------|------|------|------|------|
| GFP | 0.789 | **0.864** | **0.864** | **0.864** | **0.864** | **0.864** | **0.864** |
| TFB | 0.439 | 0.969 | **0.973** | **0.973** | 0.956 | 0.956 | 0.956 |
| UTR | 0.593 | 0.733 | **0.760** | 0.711 | 0.723 | 0.735 | 0.739 |
| SuperC | 0.399 | 0.505 | 0.524 | 0.525 | **0.526** | 0.508 | 0.501 |
| Ant | 0.565 | **0.965** | 0.962 | 0.933 | 0.947 | 0.884 | 0.945 |
| D'Kitty | 0.884 | 0.940 | **0.941** | 0.935 | 0.937 | 0.937 | 0.937 |
| Hopper | 1.000 | 1.692 | 1.989 | 2.022 | 2.149 | **2.444** | 2.112 |

Table 12: Ablation studies on BDI components considering sequential nature.

| Task | TFB | TFB(reduce) | LACT | AMP | ALIP | LEV | SUMO |
|------|------|------|------|------|------|------|------|
| BDI | **0.986** | **0.957** | **0.820** | **0.772** | **0.813** | **0.859** | **1.489** |
| w/o $\mathcal{L}_{l2h}$ | 0.954 | 0.849 | $-0.134$ | 0.765 | 0.659 | 0.577 | 0.773 |
| w/o $\mathcal{L}_{h2l}$ | 0.952 | 0.900 | 0.550 | 0.753 | 0.699 | 0.556 | 0.596 |

Table 13: Comparison between BDI and pure maximization.

| Task | GFP | TFB | UTR | SuperC | Ant | D'Kitty | Hopper |
|------|-----|-----|-----|--------|-----|---------|--------|
| Best Result | **0.865** | **0.973** | **0.760** | **0.520** | **0.962** | **0.941** | **1.989** |
| Our BDI | 0.864 | 0.973 | 0.760 | 0.520 | 0.962 | 0.941 | 1.989 |
| Pure Maxi | 0.520 | 0.911 | 0.937 | 1.830 | 0.864 | 0.956 | 0.734 |

Table 14: Experimental results on different predefined target scores (normalized).

| Task/Weight | 0.00 | 0.50 | 0.80 | 0.90 | **1.00** | 1.10 | 1.20 | 1.50 | 2.00 |
|-------------|------|------|------|------|------|------|------|------|------|
| Ant | 0.933 | 0.950 | 0.970 | 0.944 | 0.962 | 0.943 | 0.956 | 0.941 | 0.938 |
| TFB | 0.973 | 0.973 | 0.973 | 0.973 | 0.973 | 0.973 | 0.973 | 0.973 | 0.973 |

- LEV: In an ATP-dependent reaction, LG can be converted by Levoglucosan kinase to the glucose-6-phosphate. Following [57], we measure the enzyme activity of this reaction. All 7891 protein sequences are used to train the oracle and the bottom half are for the offline algorithms.

- SUMO: Following [58], we measure the growth rescue rate of human SUMO E2 conjugase protein. Around 2000 samples are used to train the oracle and the bottom half are used for the offline algorithms.

As shown in Table 12, these results demonstrate the effectiveness of our backward mapping and forward mapping. Besides the sequential nature, the structural information [59, 47] can also be considered for protein design and we leave it to future work.

## A.6  Comparison between Maximization and Regression

We change the regression to the pure maximization (Pure Maxi) of the predictions of the learned objective function and report the experimental results on all tasks.

As we can see in Table 13, Pure Maxi results are generally worse than the regression form and we conjecture that the advantage of the regression form arises due to the consistency between the forward mapping and the backward mapping which both use the same predefined target score $y_h$.

## A.7  Varying Weights

We simply assign the forward mapping and the backward mapping equal weight since the two terms are symmetric and thus viewed as equally important. In this subsection, we view the weight term of the backward mapping as a hyperparameter and study the hyperparameter sensitivity on Ant and TFB as we did in Sec 4.6.

For the ANT task, we find the results generally improve for any weight greater than zero. This demonstrates the effectiveness of backward mapping.

For the TFB task, we find the behavior of BDI is identical for different weight terms. One possible reason is that BDI here does not consider the sequential nature of DNA and thus the backward mapping cannot provide significant performance gain as we have discussed before.

## A.8  Analysis of Backward Mapping

In this subsection, we provide more visualization and analysis to verify the effectiveness of the backward mapping. We use the NTK $k(\boldsymbol{x}_i, \boldsymbol{x}_j)$ to measure the similarity between two points and compute the similarity between the generated high-scoring design $\boldsymbol{x}_h$ and the high-scoring designs $\boldsymbol{X}$ in the offline dataset. This is defined as:

$$simi(\boldsymbol{x_h}, \boldsymbol{X}) = mean(k(\boldsymbol{x}_h, \boldsymbol{X})). \tag{15}$$

As shown in Table 15, we find that $simi(\boldsymbol{x}_h^{backward}, \boldsymbol{X}) > simi(\boldsymbol{x}_h^{bdi}, \boldsymbol{X}) > simi(\boldsymbol{x}_h^{forward}, \boldsymbol{X})$. This can be explained as follows. Since the backward mapping extracts the high-scoring features from the offline dataset, $\boldsymbol{x}_h^{backward}$ is the closest to the high-scoring designs X in the offline dataset and thus $simi(\boldsymbol{x}_h^{backward}, \boldsymbol{X})$ is large. While encouraging the design $\boldsymbol{x}_h^{forward}$ to explore

Table 15: Analysis of backward mapping.

| Task | BDI | forward mapping | backward mapping |
|------|--------|-----------------|------------------|
| Ant  | 0.0752 | 0.0739          | 0.0808           |
| TFB  | 0.2646 | 0.2395          | 0.3570           |

a high score, the forward mapping alone leads to the design far from the offline dataset, and thus $simi(\boldsymbol{x}_h^{forward}, \boldsymbol{X})$ is the smallest. BDI can explore a high score (forward mapping) and stay close to the offline dataset (backward mapping), which leads to $simi(\boldsymbol{x}_h^{bdi}, \boldsymbol{X})$ between $simi(\boldsymbol{x}_h^{forward}, \boldsymbol{X})$ and $simi(\boldsymbol{x}_h^{backward}, \boldsymbol{X})$.