# OpenReview forum: "Bidirectional Learning for Offline Infinite-width Model-based Optimization"
_NeurIPS.cc/2022/Conference — NeurIPS 2022 Accept_

### Official Review · Reviewer_HR4n · 2022-07-07

**Rating:** 6
**Confidence:** 2
**Soundness:** 3 good
**Presentation:** 2 fair
**Contribution:** 3 good

**Summary:**

This paper proposes bidirectional learning for offline infinite-width model-based optimization (BDI) to address the out-of-distribution problem. BDI consists of a forward mapping and a backward mapping. The authors claim that the backward mapping can distill more information into high-scoring designs. Experiments show that BDI achieves state-of-the-art performance in both continuous and discrete tasks.

**Questions:**

Please refer to the "Strengths And Weaknesses" part.

**Limitations:**

The authors have adequately addressed the limitations and potential negative societal impact of their work.

**Strengths And Weaknesses:**

Originality:

The idea of BDI is novel. This paper introduces the idea of neural tangent kernels into offline model-based optimization. Moreover, the authors propose a novel backward mapping, significantly improving the performance of model-based optimization.

Quality:

The comparisons are thorough. Results show that BDI outperforms baselines in most tasks, and all component in BDI is useful. The use of neural tangent kernels in BDI is based on solid theoretical motivation. Nevertheless, I am confused about the backward mapping.
1. What is the motivation for using the backward mapping? The authors may want to provide an intuitive explanation of why we need a backward mapping besides the forward mapping.
2. How does the backward mapping help to distill information into the high-scoring designs? Though the authors provide ablation studies to show that the backward mapping does work, the authors may want to provide more analyses/visualization to show why the backward mapping works.
3. I can not understand the predefined score $y_h$ used in the backward mapping. In forward mapping, a high $y_h$ can encourage the algorithm to generate $X_h$ with high scores. In backward mapping, what is the purpose of training $f_\theta^h$ with a virtual score $y_h$?



Clarity:

Most of this paper is easy to follow. Nevertheless, I am confused about the backward mapping. The authors may want to discuss the backward mapping more, especially its motivation and how it works.

Significance:

The backward mapping is novel and can provide a significant performance improvement.

---

> ### Author Response · Authors · 2022-08-02
> **Response to review questions 1/n**
>
> ### General Reply
>
> Many thanks for your valuable and constructive comments on clarifying, correcting, and improving the materials in this paper! We will carefully revise the paper according to your comments as explained below.
>
> ### Strengths And Weaknesses:
>
>
>     > 1. What is the motivation for using the backward mapping? The authors may want to provide
>     an intuitive explanation of why we need a backward mapping besides the forward mapping.
>
>
> The backward mapping is inspired by data distillation [4] and the Decision Transformer [3]. As we discuss in L329-L335 of the related work section, data distillation [4] keeps the model fixed and attempts to distill the knowledge from a large training dataset into a small one. The experiments in [4] are on MNIST (and CIFAR) with  10 (100 for CIFAR) distilled images with predefined labels being digits in the range 0-9 (categories such as dog, cat...for CIFAR). We observe that the distilled images extract the features of the corresponding digit (dog, cat...for CIFAR) from the training dataset.
>
>     [3] Chen et al. Decision transformer: reinforcement learning via sequence modeling. Proc. Adv. Neur. Inf. Proc. Syst, 2021.
>     [4] Wang et al. Dataset distillation. arXiv preprint, 2018.
>
> This observation motivates us to consider what the distilled sample could be if the label becomes a large virtual predefined target score yh in the offline model-based optimization setting. We compare data distillation and backward mapping in the following table, which should help explain our motivation.
>
> |  Comparison  | Data distillation [4]  |    Backward mapping  |
> |:----:|:----:|:----:|
> | Data x | image in the training dataset D | design in the offline dataset D |
> | Label type y_t | 0-9 for MNIST (dog, cat...for CIFAR); | some measurement of protein/dna/robot...  |
> | Label value y_v | within the training data | larger than the max of the offline dataset  |
> | Task type | classification | regression  |
> | Task objective | distill $\tilde{x}$ to predict D  | distill xh to incorporate high-scoring features of  D |
>
> The introduction of the large virtual predefined target score yh is inspired by Decision Transformer, in which a reward (score) is predefined for reinforcement learning, as we discuss in L64-L66.
>
> Forward mapping alone suffers from the out-of-distribution issue as we discuss in L34-L37. We will add the above table and the following sentences  to the paper to address the suggestion "to provide an intuitive explanation of why we need a backward mapping besides the forward mapping":
>
>     Forward mapping alone suffers from the out-of-distribution issue. The backward mapping leverages the high-scoring design to predict the offline dataset and thus distills the information of the offline dataset into the high-scoring design, which can mitigate the out-of-distribution issue.

---

> > ### Author Response · Authors · 2022-08-02
> > **Response to review questions 2/n**
> >
> >     > 2. How does the backward mapping help to distill information into the high-scoring designs?
> >     Though the authors provide ablation studies to show that the backward mapping does work,
> >     the authors may want to provide more analyses/visualization to show why the backward mapping works.
> >
> > We discuss the motivation of the backward mapping in L34-L57. As shown in Figure 1, pure gradient ascent on the proxy function yields the seemingly high-scoring design $p_{grad}$ with a low ground-truth score. From a model perspective, previous work attempted to train a better proxy closer to the ground-truth to mitigate the out-of-distribution issue. Instead, from a data perspective, together with the forward mapping, our backward mapping can align the high-scoring design with the offline dataset, which constrains the high-scoring design to distill more information from the offline dataset. This leads to a better high-scoring design.
> >
> > We discuss the inner mathematical mechanism in L163-L168 **Analysis of M=1**.
> > In our experiments, we aim for one high-scoring design $M=1$ and by minimizing Eq.(13) we can find a high-scoring design xh such that any design in the offline dataset similar to the high scoring design is encouraged to have a high score prediction. In this way, xh tries to incorporate as many high-scoring features from the offline dataset as possible. For the M=1 case, we can observe that a good xh will both (1) have a high predicted score based on the forward mapping; and (2) be able to predict (relatively) high scores for multiple (relatively) high scoring designs in the offline dataset based on backward mapping.
> >
> > As per your suggestion, we also provide additional analysis to verify and explain the effectiveness of the backward mapping. We use the NTK k(. , .) to measure the similarity between two points and compute the similarity between the generated high-scoring design xh and the high-scoring designs X in the offline dataset. This is defined as: simi(xh, X) = mean(k(xh, X)). We report the results in the table below.  We have added this part into Appendix A.8.
> >
> > |  Task  | BDI  | forward map  | backward map
> > |:----:|:----:|:----:|:----:|
> > |Ant | $0.0752$   | $0.0739$  |  $0.0808$ |
> > |TFB | $0.2646$   | $0.2395$  |  $0.3570$ |
> >
> > We find that simi(xhback, X) > simi(xhbdi, X) > simi(xhforward, X). This can be explained as follows. Since the backward mapping extracts the high-scoring features from the offline dataset, xhback is the closest to the high-scoring designs X in the offline dataset and thus simi(xhbdi, X) is large. While encouraging the design xhforward to explore a high score, the forward mapping alone leads to a design that is  far from the offline dataset and thus simi(xhforward, X) is the smallest. BDI can explore a high score (forward mapping) and stay close to the offline dataset (backward mapping), which leads to simi(xhbdi, X) between simi(xhforward, X) and simi(xhback, X).
> >
> >     > 3. I can not understand the predefined score  used in the backward mapping.
> >     In forward mapping, a high  can encourage the algorithm to generate  with high scores.
> >     In backward mapping, what is the purpose of training with a virtual score ?
> >
> >
> > The formulation of BDI effectively supposes that there is a design xh with the predefined target score yh. It aims to identify the best xh so that using the NTK: (a) the offline dataset can predict the predefined target score yh for the selected xh (forward mapping); and (b) the selected xh, with associated label yh, can predict the labels of (relatively) high scoring designs in the offline dataset (backward mapping). The yh value thus plays a role in both forward and backward mapping; our experiments demonstrate that performance is robust to the specific choice of yh.
> >
> >
> >
> > ### Overall
> >
> > **Does the above reply address your concerns? Thank you again for your instructive review and thoughtful feedback. We hope that there is the opportunity for further discussion with you during the rebuttal phase.**

---

> > > ### Author Response · Authors · 2022-08-08
> > > **Kindly Request for Feedback**
> > >
> > > Thanks for your detailed review and the questions you raised. We tried to thoroughly elaborate on your raised questions regarding the motivation of backward mapping. Additionally, we have also conducted more experiments to explain the effectiveness of backward mapping.
> > >
> > > Could you please kindly provide us with more feedback if we did not fully address your concerns? Thank you and looking forward to further exchanging our thoughts with you.

---

> > > > ### Comment · Reviewer_HR4n · 2022-08-08
> > > > **Thank you for the response**
> > > >
> > > > Thank you for the response. The response has properly addressed my concerns. I will increase my score.

---

> > > > > ### Author Response · Authors · 2022-08-08
> > > > > **Thank you**
> > > > >
> > > > > Thank you again for your constructive feedback and appreciation.
> > > > >
> > > > > We will certainly include our main points of the above response in an improved version.

---

### Official Review · Reviewer_qG1d · 2022-07-11

**Rating:** 7
**Confidence:** 4
**Soundness:** 3 good
**Presentation:** 3 good
**Contribution:** 3 good

**Summary:**

Offline model-based optimization could suffer from the distributional shift issue where the model bias could be exploited by the optimization process to output poor designs. Most previous offline model-based optimization aims to solve the problem by introducing loss function terms that promote distribution matching between the learned model outputs and the ground-truth. Unlike previous works, this paper introduces a directional learning approach based on neural tangent kernel to facilitate the distribution matching. By encouraging learning accurate mappings from low-scoring designs to high-scoring designs and vice versa, the proposed method could alleviate the issue of outputting unseen designs with significantly overestimated scores. The proposed method achieved the best performance on six out of seven widely studied offline model-based optimization task.

**Questions:**

Could the authors explain why using a constant $y_h$ could work? Would it be possible to further improve the proposed method with a more sophisticated design of $y_h$ where the values are different for each high-scoring sample?

**Limitations:**

The authors have addressed the limitations.

**Strengths And Weaknesses:**

### Strengths
1. The proposed approach is well motivated and the writing is easy to follow.
2. The authors provide a good insight on why the bidirectional learning helps mitigate the distributional shift issue in 3.3.
3. Experiment results show strong performance of the proposed method.

### Weakness
This paper lacks a discussion on the relationship between the directional learning and a conservative regularization. I believe that adding the backward mapping loss is similar to a conservative regularizer in the sense that they both aim to alleviate the distributional shift between the predicted optimal designs and the dataset designs. I hope that the authors could provide some discussions on their similarity/difference and why learning the backward mapping could be a better alternative than conservative regularization methods.

---

> ### Author Response · Authors · 2022-08-02
> **Response to review questions**
>
> ### General Reply
>
> Many thanks for your valuable and constructive comments on clarifying, correcting, and improving the materials in this paper! We will carefully revise the paper according to your comments as explained below.
>
> ### Strengths And Weaknesses
>
>     >This paper lacks a discussion on the relationship between the directional learning and a conservative regularization.
>     >I believe that adding the backward mapping loss is similar to a conservative regularizer in the sense that they both aim to alleviate the distributional shift between the predicted optimal designs and the dataset designs.
>     > I hope that the authors could provide some discussions on their similarity/difference and why learning the backward mapping could be a better alternative than conservative regularization methods.
>
>
> #### Compare with conservative regularization
>
> Thank you for pointing this out! We agree that both backward mapping and the regularization term in COMs can mitigate the out-of-distribution issue. The key difference is that COMs achieves this from a model perspective while BDI achieves this from a data perspective.
>
> Compared with COMs, the backward mapping in BDI has two advantages:
>
> 1. Effectiveness: Real-world offline model-based optimization often involves small-scale datasets since the labeling cost of protein/dna/robot is very high. BDI is built on the neural tangent kernel, which generally performs better than finite neural networks on small-scale datasets [1]. COMs is built on a finite neural network and it is non-trivial to modify COMs to a neural tangent kernel version.
>
> 2. Generalization: The infinitely wide deep neural network (the neural tangent kernel) represents a broad class of DNNs [2], which enhances the generalization of the high-scoring designs. It is non-trivial to apply the infinitely wide DNN in COMs.
>
>     [1] Arora et al. Harnessing the power of infinitely wide deep nets on small-data tasks. Int. Conf. Learning Representations, 2020.
>
>     [2] Yuan et al. Neural tangent generalization attacks. Int. Conf. on Machine Learning, 2021.
>
> We will add this part to the section of experiments.
>
>
> ### Questions
>
> #### Constant y_h
>
>     >Could the authors explain why using a constant y_h could work? Would it be possible to
>     further improve the proposed method with a more sophisticated design of  where the values are
>     different for each high-scoring sample?
>
> We set the constant $y_h$ to be larger than the maximal value of the offline dataset and thus the corresponding design can extract the high-scoring features like we analyze in L163-L168.
>
> We investigate this further in Appendix A.2 (b), where we select 1) multiple relatively high-scoring ground-truth designs from the offline dataset; and 2) one learnable high-scoring design with the predefined target score yh=10 to form the set of high-scoring designs. In this case, different high-scoring samples have different yh. This setting also yields good results.
>
> Thank you for the suggestion regarding different yh. We believe it would be possible to further improve the proposed method with a more sophisticated design in which the values are different for each high-scoring sample. We could strive to optimize over the values of yh (constraining them to be above a threshold). This is more complicated than the current approach, but it is a valuable suggestion for future work, and we will add a comment to this effect in the paper.
>
> ### Overall
>
> **Thank you again for your instructive feedback on our paper. Please let us know if we have resolved your concern or if you have any further questions.**

---

> > ### Comment · Reviewer_qG1d · 2022-08-07
> > **.**
> >
> > I'd like to thank the authors for their response. I will keep my rating of 7.

---

> > > ### Author Response · Authors · 2022-08-08
> > > **Thank you**
> > >
> > > Thank you again for your constructive feedback. As we mentioned in the previous response, we will include your valuable suggestion in an improved draft.

---

### Official Review · Reviewer_QNRb · 2022-07-12

**Rating:** 5
**Confidence:** 4
**Soundness:** 3 good
**Presentation:** 3 good
**Contribution:** 3 good

**Summary:**

This paper investigates the subject of Offline Model-Based Optimization, which involves processing a static dataset of designs X and function evaluations Y in order to solve the optimization problem X* = \argmax_{X} f(X) for an unknown function f. This paper proposes a gradient-based method for solving Offline MBO problems using infinite-width DNN models. The approach facilitates efficiently optimizing designs X via a proposed bidirectional objective that encourages designs to be found that not only achieve high performance under an approximation of the objective function f_{\theta} (X), which is referred to as the forward mapping in the paper, but jointly maximize a distillation objective that encourages optimized design-score pairs to be informative about the characteristics of the true objective function (the backward mapping).

One key novelty in the paper is the proposed use of a distillation objective to constrain the optimized design-score pairs, rather than regularizing the DNN model used for optimization. Results in the paper appear to show the effectiveness of this principle on a standard benchmarking framework for Offline MBO using a comprehensive set of baselines.

**Questions:**

(1) There appears to be a discrepancy between a small fraction of the results presented in the main performance table, and those reported in other recent papers. In particular, the performance of MIN and COMs are different from those reported in design-bench (Trabucco, B. et al., 2022) on the following tasks: Ant Morphology, DKitty Morphology, and Hopper Controller. Could the authors investigate this and explain the cause of this difference?

(2) The optimization objective in Equation (4) is chosen as a regression objective, rather than purely maximizing the predictions of the learned objective function f_{\theta} (X). How are the results affected when this equation is changed to pure maximization, rather than regression?

(3) The objective functions in Equation (8) appear to be assigned equal weight. How are the results affected by ablating the weight of the distillation term (Equation 6) in the loss function?

**Limitations:**

The authors have sufficiently addressed these.

**Strengths And Weaknesses:**

Originality:

The subject of mitigating out-of-distribution samples (ie, design-score pairs) using gradient-based Offline Model-Based Optimization techniques has been the subject of multiple papers. The subject itself is not novel, but the solution proposed by the authors (regularizing the optimized samples rather than the model) is an original proposition in the Offline MBO domain. Though different in execution, a similarly inspired technique has been proposed in the Offline RL setting, namely BCQ (Fujimoto, S. et al., 2019), which constrains the (s’, a’) pairs sampled during bellman backups \max_{a’} Q(s’, a’) to be contained in the support of the static dataset.

Both papers explore constraints (either implicitly or explicitly) on the data being optimized (actions in the case of BCQ, and designs in the case of BDI), rather than model regularization. However, it should be noted that the implementation of the constraint is fundamentally different between these two approaches, and BDI’s implementation is original. Dataset distillation enforces a constraint on the data that is an appealing alternative to the support-based constraint seen in BCQ because it lets the model generate designs outside the training distribution.


Quality:


The experimental setup in the paper adheres to the evaluation protocol established in prior benchmarking papers, which aids in the reproducibility of the paper. Relevant baselines from the literature, including several new methods, and multiple classical methods, adapted to the Offline MBO setting, are reported. These factors improve the quality of the paper’s main evaluation. Certain relevant ablations are not included (see my questions below), and a small number of claims and speculations made in the paper are not fully supported by evidence. For example, on line 278-279 of the manuscript, the authors write “we observe that the backward mapping is more effective for continuous tasks, possibly because we do not model the sequential nature of DNA and protein sequences.” This is a helpful intuition, and could be improved if the authors tested this hypothesis with a model that does “model the sequential nature of DNA and protein sequences.” Quality may also be improved by addressing the questions listed below.

Clarity:


Certain experimental details are not clear from the manuscript. For example, on line 194-195 the authors write “we choose the top N = 128 most promising designs for each method,” which could suggest more than N = 128 designs are generated, and the number is downsampled to 128 using some indicator for how promising the designs are. If such a post-selection mechanism is used, additional details should be included in the paper for reproducibility. Such a mechanism should query the ground truth objective function only N = 128 times to compute 100th percentile and 50th percentile statistics as per design-bench (Trabucco, B. et al., 2022).

Significance:


Mitigating the out-of-distribution problem in Offline MBO has significance as a research area, due to its compelling applications in real-world design problems. The results in this paper show the proposed approach is quite effective in practice at finding high-performing designs.

---

> ### Author Response · Authors · 2022-08-02
> **Response to review questions 1/n**
>
> ### General Reply
>
> Many thanks for your valuable and constructive comments on clarifying, correcting, and improving the materials in this paper! We will carefully revise the paper according to your comments as explained below.
>
> ### Originality
>
> Thank you for drawing our attention to the interesting BCQ work which used regularization on the optimized samples in the offline RL setting. We will add the following sentences in the related work section to acknowledge the connection:
>
>     > Similar to our backward mapping constraining designs, BCQ (Fujimoto, S. et al., 2019)
>     > constrains the state-action pairs to be contained in the support of the static dataset.
>     > Our proposed backward mapping is different in its implementation and also enables the model to generate designs outside the training distribution.

---

> > ### Author Response · Authors · 2022-08-02
> > **Response to review questions 2/n**
> >
> > ### Quality
> >
> >     > For example, on line 278-279 of the manuscript, the authors write “we observe
> >     > that the backward mapping is more effective for continuous tasks, possibly because we do not
> >     > model the sequential nature of DNA and protein sequences.” This is a helpful intuition,
> >     > and could be improved if the authors tested this hypothesis with a model
> >     > that does “model the sequential nature of DNA and protein sequences.”
> >     > Quality may also be improved by addressing the questions listed below.
> >
> > Thanks for your suggestions! We conduct additional experiments to verify our intuition in L278-L279.
> >
> > We adopt DNABert [1] and ProtBert [2]  to model the sequential nature of DNA/protein. We conduct an ablation study, removing the backward mapping to verify its effectiveness. We also remove the forward mapping to assess its importance. The following experiments demonstrate our intuition: backward mapping matters. We have added this part into Appendix A.5.
> >
> > For DNA experiments, we consider the task with an exact oracle: TFBind8, and obtain the following 100th percentile results. As we can see in the following table, both the backward mapping and the forward mapping matter in BDI. We further consider a harder task setting TFBind8(reduce): reduce the size of the offline dataset from 32896 to 5000. The results verify the effectiveness and robustness of bidirectional mappings.
> >
> > |  DNA Task  | BDI  | w/o $\mathcal{L}_{l2h}$   | w/o $\mathcal{L}_{h2l}$
> > |:----:|:----:|:----:|:----:|
> > |TFBind8 | **0.986**   | $0.954$  |  $0.952$ |
> > |TFBind8(reduce) | **0.957**   | $0.849$  |  $0.900$ |
> >
> > For protein experiments, we first consider the GFP task we used in our paper, and obtain the following results:
> >
> > |  Protein Task  | BDI  | w/o $\mathcal{L}_{l2h}$   | w/o $\mathcal{L}_{h2l}$
> > |:----:|:----:|:----:|:----:|
> > |GFP | **0.864**   | $0.864$  |  $0.864$ |
> >
> > The results are indistinguishable. We note that [3] observes for the GFP task "indistinguishable results across all methods". To further investigate the behavior, we introduce $5$ new protein tasks used in recent works: LACT, AMP, ALIP, LEV, and SUMO. We now present the results and task details.
> >
> > |  Protein Task  | BDI  | w/o $\mathcal{L}_{l2h}$   | w/o $\mathcal{L}_{h2l}$
> > |:----:|:----:|:----:|:----:|
> > | LACT | **0.820** | $-0.134$ |  $0.550$ |
> > | AMP | **0.772**   | $0.765$ |  $0.753$ |
> > | ALIP | **0.813**   | $0.659$  | $0.699$ |
> > | LEV | **0.859**   | $0.577$ |  $0.556$ |
> > | SUMO | **1.489**   | $0.773$  | $0.596$ |
> >
> > These results demonstrate the effectiveness of our bidirectional mappings.
> >
> > LACT: We measure the thermodynamic stability of the TEM-1 β-Lactamase protein. The oracle is trained on $17857$ samples from [4][5]. The bottom half of these samples are used for offline algorithms.
> >
> > AMP: Antimicrobial peptides are short protein sequences that act against pathogens. We use the $6760$ AMPs from [6] to train the oracle and the bottom half of the AMPs for the offline algorithms.
> >
> > ALIP: We measure the enzyme activity of the Aliphatic Amide Hydrolase sequence following [7]. We use the $6629$ samples from [7] to train the oracle and the bottom half of them for the offline algorithms.
> >
> > LEV: In an ATP-dependent reaction, LG can be converted by Levoglucosan kinase to the glucose-6-phosphate. Following [8], we measure the enzyme activity of this reaction. All $7891$ protein sequences are used to train the oracle and the bottom half are for the offline algorithms.
> >
> > SUMO: Following [9], we measure the growth rescue rate of human SUMO E2 conjugase protein. Around 2000 samples are used to train the oracle and the bottom half are used for the offline algorithms.
> >
> >     [1] Ji et al. DNABERT: pre-trained bidirectional encoder representations from transformers model for DNA-language in genome. Bioinformatics, 2021.
> >     [2] Ahmed et al. ProtTrans: towards cracking the language of lifes code through self-supervised deep learning and high performance computing. IEEE Transactions on Pattern Analysis and Machine Intelligence, 2020.
> >     [3] Trabucco et al. Design-bench: benchmarks for data-driven offline model-based optimization. arXiv preprint, 2022.
> >     [4] CE et al. Pervasive  pairwise intragenic epistasis among sequential mutations in tem1 β-lactamase. Journal of Molecular Biology, 2019.
> >     [5] Firnberg et al. A comprehensive, high-resolution map of a gene’s fitness landscape. Molecular Biology and Evolution, 2014.
> >     [6] Zhang et al. Unifying likelihood-free inference with black-box optimization and beyond. Int. Conf. Learning Representations, 2022.
> >     [7] Wrenbeck et al. Single mutation fitness landscapes for an enzyme on multiple substrates reveal specificity is globally encoded. Nature Communications, 2017.
> >     [8] Klesmith et al. Comprehensive sequence-flux mapping of a levoglucosan utilization pathway in e.coli. ACS Synthetic Biology, 2015.
> >     [9] Weile et al. A framework for exhaustively mapping functional missense variants. Molecular Systems Biology, 2017.

---

> > > ### Author Response · Authors · 2022-08-02
> > > **Response to review questions 3/n**
> > >
> > > ### Clarity
> > >
> > >     > Certain experimental details are not clear from the manuscript.
> > >     > For example, on line 194-195 the authors write “we choose the top N = 128 most promising
> > >     > designs for each method,” which could suggest more than N = 128 designs are generated,
> > >     > and the number is downsampled to 128 using some indicator for how promising the designs are.
> > >     > If such a post-selection mechanism is used, additional details should be included
> > >     > in the paper for reproducibility. Such a mechanism should query the ground truth
> > >     > objective function only N = 128 times to compute 100th percentile
> > >     > and 50th percentile statistics as per design-bench (Trabucco, B. et al., 2022).
> > >
> > >
> > > We apologize for the ambiguous wording. We follow exactly the same procedure as presented in Design-bench (Trabucco, B. et al., 2022) [10], and we only generate the top $128$ candidates designs. We will clarify this part in our final version to avoid possible misunderstanding.
> > >
> > >
> > >     [10] Trabucco et al. Conservative objective  models for effective offline model-based optimization. Int. Conf. Learning Rep, 2021.

---

> > > > ### Author Response · Authors · 2022-08-02
> > > > **Response to review questions 4/n**
> > > >
> > > > ### Questions
> > > >
> > > >     > (1) There appears to be a discrepancy between a small fraction of the results presented
> > > >     > in the main performance table, and those reported in other recent papers.
> > > >     > In particular, the performance of MIN and COMs are different from those reported in design-bench
> > > >     > (Trabucco, B. et al., 2022) on the following tasks: Ant Morphology, DKitty Morphology, and
> > > >     > Hopper Controller.
> > > >     > Could the authors investigate this and explain the cause of this difference?
> > > >
> > > >
> > > > #### 1. Small discrepancy between ours and previous published results
> > > > For the baseline MIN, we run the code released in  (Trabucco, B. et al., 2022) and we conjecture that the small difference arises due to different random seeds and/or different machines.
> > > >
> > > > The baseline COMs belongs to the gradient updating category. For all methods in this category, the time step $T$ is set as $200$ in our work but as $50$ in (Trabucco, B. et al., 2022). We set $T$ to be large following ROMA [11] "set the number of solution update to be large enough". Other factors like neural network architectures can also lead to result differences. Compared with the COMs results reported in (Trabucco, B. et al., 2022), our BDI achieves better results on four tasks and the same result on the GFP task. The results for the D’Kitty Morphology task and the Hopper Controller task are slightly worse. Using the reported results in (Trabucco, B. et al., 2022), our BDI is still the best performing method and outperforms COMs in terms rank mean(2.1/11 < 3.9/11) and rank median (2/11 < 3/11).
> > > >
> > > > We will report the original performances of MIN and COMs at (Trabucco, B. et al., 2022) in our final version.
> > > >
> > > >
> > > > It is worth emphasizing that there is no validation set in offline model-based optimization (we are searching for designs outside the training set and in practice do not have access to a performance oracle) so we cannot adopt standard approaches for determining hyperparameters [12] such as the number of gradient steps.
> > > >
> > > >     [11] Yu et al. Roma: Robust model adaptation for offline model-based optimization. NeurIPS, 2021.
> > > >     [12] Trabucco et al. Design-bench: benchmarks for data-driven offline model-based optimization.  arXiv preprint, 2022.
> > > >
> > > >
> > > >     > (2) The optimization objective in Equation (4) is chosen as a regression objective,
> > > >     > rather than purely maximizing the predictions of the learned objective function f_{\theta} (X).
> > > >     > How are the results affected when this equation is changed to pure maximization,
> > > >     > rather than regression?
> > > >
> > > >
> > > > #### 2. Change to pure maximization
> > > >
> > > > We change the regression to pure maximization (Pure Maxi) the predictions of the learned objective function and report the experimental results on all tasks.  We have added this part into Appendix A.6.
> > > >
> > > > |  Mode/Task  | SuperC  | Ant   | D'Kitty  | Hopper   | GFP  | TFB   | UTR  |
> > > > |:----:|:----:|:----:|:----:|:----:|:----:|:----:|:----:|
> > > > | Our BDI | $0.520$ | $0.962$ | $0.941$ | $1.989$ | $0.864$ | $0.973$ | $0.760$ |
> > > > | Pure Maxi | $0.520$ | $0.911$ | $0.937$ | $1.830$ | $0.864$ | $0.956$ | $0.734$ |
> > > > | Best Baseline | $0.503$ | $1.214$ | $0.942$ | $1.959$ | $0.865$ | $0.953$ | $0.707$ |
> > > > | Best Result | **0.520** | **1.214** | **0.942** | **1.989** | **0.865** | **0.973** | **0.760** |
> > > >
> > > >
> > > > As we can see, the Pure Maxi results are generally worse than the regression form. We conjecture that the advantage of the regression form arises due to the consistency between the forward mapping and the backward mapping since both use the same predefined target score yh.
> > > >
> > > >     > (3) The objective functions in Equation (8) appear to be assigned equal weight.
> > > >     > How are the results affected by ablating the weight of the distillation term (Equation 6) in the loss function?
> > > >
> > > >
> > > > #### 3. Equal weight
> > > >
> > > > We simply assign the forward mapping and the backward mapping equal weight since the two terms are symmetric and thus viewed as equally important. To answer your question, we view the weight term of the backward mapping as a hyperparameter and study the hyperparameter sensitivity on Ant and TFB like we did in Sec 4.6. We have added this part into Appendix A.7.
> > > >
> > > > |  Weight  | Ant  | TFB   |
> > > > |:----:|:----:|:----:|
> > > > | $0.00$ | $0.933$ | $0.973$ |
> > > > | $0.50$ | $0.950$ | $0.973$  |
> > > > | $0.80$ | $0.970$ | $0.973$  |
> > > > | $0.90$ | $0.944$ | $0.973$  |
> > > > | **1.00** | $0.962$ | $0.973$ |
> > > > | $1.10$ | $0.943$ | $0.973$  |
> > > > | $1.20$ | $0.956$ | $0.973$  |
> > > > | $1.50$ | $0.941$ | $0.973$  |
> > > > | $2.00$ | $0.938$ | $0.973$  |
> > > >
> > > > For the ANT task, we find the results generally improve for any weight greater than zero. This demonstrates the effectiveness of the backward mapping.
> > > >
> > > > For the TFB task, we find the behavior of BDI is identical for different weight terms. One possible reason is that BDI here does not consider the sequential nature of DNA and thus the backward mapping cannot provide significant performance gain as we have discussed before.

---

> > > > > ### Author Response · Authors · 2022-08-02
> > > > > **Response to review questions 5/n**
> > > > >
> > > > > ### Overall
> > > > >
> > > > > **Does the above reply address your concerns? Thank you again for your instructive review and feedback. We very much appreciate your careful review of the paper and look forward to further exchange with you during the rebuttal phase.**

---

> > > > > > ### Author Response · Authors · 2022-08-08
> > > > > > **Kindly Request for Feedback**
> > > > > >
> > > > > > Thank you for your detailed review and your constructive feedback. We tried to address all your listed concerns above, mainly including:
> > > > > >
> > > > > > - We have discussed the relationship of our work to one of the related work of BCQ;
> > > > > > - We have further investigated the effectiveness of backward mapping when considering the sequential nature;
> > > > > > - We have clarified our evaluation setting;
> > > > > > - We have explained the reason behind the small discrepancy between ours and previously published results (MIN and COMs);
> > > > > > - We have reported the results after changing regression to pure maximization;
> > > > > > - We have conducted the ablation study for the weight of the distillation term.
> > > > > >
> > > > > > Could you please further elaborate if anything is still not clear? Thank you and looking forward for your feedback.

---

> > > > > > > ### Comment · Reviewer_QNRb · 2022-08-09
> > > > > > > **Great response, questions are resolved**
> > > > > > >
> > > > > > > Thanks for responding to my questions at length, each of my major concerns has been resolved. In my current evaluation of the paper, I intend to increase my rating, pending further discussion with the other reviewers and the Area Chair.

---

> > > > > > > > ### Author Response · Authors · 2022-08-09
> > > > > > > > **Thank you**
> > > > > > > >
> > > > > > > > Thank you for your great questions, which make our paper stronger.
> > > > > > > >
> > > > > > > > We will certainly include the main points of our discussion in an improved version.

---

### Meta-Review · Area_Chair_YmD2 · 2022-09-12

**Recommendation:** Accept
**Confidence:** Certain

**Metareview:**

This paper studies Offline Model-Based Optimization. This paper proposes a gradient-based method for solving Offline MBO problems using infinite-width Deep learning models.
The key novelty of the paper is in proposed use of a distillation objective to constrain the optimized design-score pairs.

All three reviewers identify the novelty of the problem and the approach.
The paper also presents strong empirical evaluation on standard benchmarks.

The rebuttal discusses yielded constructive changes in the paper, and the authors are expected to account for the discussion and suggestions in the next iteration of the manuscipt.
The AC concurs with the reviews and the discussion thereafter.

**Award:**

No

---

### Decision · Program_Chairs · 2022-09-14

Accept